# Development of a model-inference system for estimating epidemiological characteristics of SARS-CoV-2 variants of concern

Wan Yang [1]✉ & Jeffrey Shaman [2]✉

To support COVID-19 pandemic planning, we develop a model-inference system to estimate epidemiological properties of new SARS-CoV-2 variants of concern using case and mortality data while accounting for under-ascertainment, disease seasonality, non-pharmaceutical interventions, and mass-vaccination. Applying this system to study three variants of concern, we estimate that B.1.1.7 has a 46.6% (95% CI: 32.3–54.6%) transmissibility increase but nominal immune escape from protection induced by prior wild-type infection; B.1.351 has a 32.4% (95% CI: 14.6–48.0%) transmissibility increase and 61.3% (95% CI: 42.6–85.8%) immune escape; and P.1 has a 43.3% (95% CI: 30.3–65.3%) transmissibility increase and 52.5% (95% CI: 0–75.8%) immune escape. Model simulations indicate that B.1.351 and P.1 could outcompete B.1.1.7 and lead to increased infections. Our findings highlight the importance of preventing the spread of variants of concern, via continued preventive measures, prompt mass-vaccination, continued vaccine efficacy monitoring, and possible updating of vaccine formulations to ensure high efficacy.

[1] Department of Epidemiology, Mailman School of Public Health, Columbia University, New York, NY, USA. [2] Department of Environmental Health Sciences, Mailman School of Public Health, Columbia University, New York, NY, USA. ✉email: wy2202@cumc.columbia.edu; jls106@cumc.columbia.edu

Multiple SARS-CoV-2 variants have been identified since summer 2020. Among these, three variants—namely, B.1.1.7, B.1.351, and P.1—have been classified as variants of concern (VOCs), per evidence indicating these genotypes are substantially more transmissible, evade prior immunity (either vaccine-induced or conferred by natural infection with wild-type virus), increase disease severity, reduce the effectiveness of treatments or vaccines, or cause diagnostic detection failures[1,2]. Multiple lines of evidence indicate the B.1.1.7 variant is roughly 50% more transmissible than wild-type virus but does not produce antigenic escape[3–6]. Further, several studies have shown that both the B.1.351 and P.1 variants are resistant to neutralization by convalescent plasma from individuals previously infected by wild-type SARS-CoV-2 viruses and sera from vaccinated individuals[5–9]; however, changes to the transmissibility of these latter two variants are less well resolved. A better understanding of the transmissibility and immune escape properties of VOCs such as B.1.351 and P.1 is needed to anticipate future COVID-19 pandemic outcomes and support public health planning.

In this study, we develop a model-inference system to estimate the relative change in transmissibility and level of immune evasion for different SARS-CoV-2 variants, while accounting for under-detection of infection, delays of reporting, disease seasonality, non-pharmaceutical interventions (NPIs), and vaccination. Testing using model-generated synthetic incidence and mortality data indicates this inference system is able to accurately identify shifts in transmissibility and immune evasion. We then apply the validated inference system in conjunction with incidence and mortality data from the UK, South Africa, and Brazil—the three countries where VOCs B.1.1.7, B.1.351, and P.1 were first identified, respectively—to estimate the change of transmissibility and immune evasion for the three VOCs, separately. We further use these inferred findings in a multi-variant, age-structured model to simulate epidemic outcomes in a municipality like New York City (NYC) where multiple variants, including B.1.1.7, B.1.351, and P.1, have been detected.

## Results

**The model-inference system and validation.** We first tested our model-inference system using 10 model-generated synthetic datasets, depicting different combinations of population susceptibility prior to the emergence of a new variant, changes in transmissibility and immune evasion for the new variant, and infection-detection rate. As population susceptibility, interventions, and disease seasonality can all affect apparent transmissibility at a given time and in order to focus on variant-specific properties, here we defined transmissibility as the average number of secondary infections per primary infection, after removing the effects of these three factors (see "Methods" section). We then quantified the change in transmissibility as its relative increase once the new variant becomes dominant. Similarly, we quantify the level of immune evasion as the increase in susceptibility after the new variant becomes dominant, relative to prior population immunity from wild-type infection.

Figure 1 shows example test results comparing model-inference system estimates with model-generated "true" values of transmissibility and susceptibility using an infection-detection rate of 20%. Across a range of epidemic dynamics, the model-inference system is able to fit both weekly incidence and mortality data (Fig. 1a) and estimate the transmissibility and susceptibility over time for both the initial pandemic wave and the subsequent pandemic wave caused by a new variant (Fig. 1b). In addition, when aggregated over both pandemic waves, the model-inference system is able to estimate the relative changes in transmissibility and immune evasion due to a new variant (Fig. 1c). When the

system is less well constrained (e.g., an infection-detection rate of 10%), model estimates, albeit less accurate, still closely track true values in most instances (Supplementary Fig. 1).

**Reconstructed pandemic dynamics in the three countries.** Following the initial emergence of SARS-CoV-2 in early 2020, the UK, South Africa, and Brazil experienced very different epidemics (Fig. 2). The model-inference system is able to recreate the observed incidence and mortality epidemic curves for all three countries (Fig. 2a, c, e). Further, cross-validation using independent data shows that the model estimates closely match measured SARS-CoV-2 prevalence in the UK[10,11] and serology measures of cumulative infection rates in South Africa[12,13] and Brazil[14], respectively (Fig. 2b, d, f). These results indicate the model-inference system accurately estimates the underlying transmission dynamics for all three countries.

In the UK, a prompt lockdown allowed the country to contain the first pandemic wave (Fig. 3). The real-time effective reproduction number ($R_t$), which measures the average number of secondary infections at a given point in time, dropped from 2.2 (95% CrI: 1.0–3.9) during the week of 3/1/2020 to below 1 during the week of 3/22/2020, the first week of the lockdown (Fig. 3a). By the week of 6/28/2020 (the week with the lowest incidence following the first pandemic wave), 6.4% (95% CrI: 3.6–12.3%) of the UK population are estimated to have been infected. However, with the relaxation of intervention measures during the summer, the transmission gradually increased again (as indicated by the estimated $R_t > 1$), leading to a large surge of infections in the autumn of 2020 (Figs. 2a, b and 3a). A second lockdown implemented in Nov 2020 reduced transmission transiently ($R_t$ was below 1 during the 4-week lockdown period; Fig. 3a). Shortly thereafter widespread transmission of the B.1.1.7 variant led to a further increase of cases before this activity was curtailed by a third lockdown and mass-vaccination.

In South Africa, the initial transmission was low likely due to a strong public health response (a lockdown was implemented from 3/26/20 to 4/16/20) and less conducive conditions for transmission during southern hemisphere summer and autumn (Figs. 2c, 3d, and Supplementary Fig. 2b). However, as the country relaxed intervention measures and entered the winter, transmission increased substantially from May 2020 onwards, leading to a large pandemic wave during May–Sep 2020. After accounting for under-detection of infection (Supplementary Fig. 3c), consistent with serology data[12,13], the model-inference system estimates that 30.0% (95% CrI: 18.0–47.1%) of the population had been infected by the week of 9/20/2020 (i.e., the week with the lowest incidence following the first pandemic wave; Fig. 2d). After two months with relatively low incidence, the emergence of the B.1.351 variant led to a resurgence of infections in late 2020 and a larger second pandemic wave (Fig. 3d). By the week of 4/11/21, another 38.6% (95% CrI: 24.1–61.1%) of the population are estimated to have been infected, including re-infections.

In Brazil, no national lockdown was implemented during the pandemic. A large first pandemic wave occurred during Mar–Oct 2020 (Figs. 2e and 3g). By the week of 11/1/2020 (i.e., the week with the lowest incidence following the first wave), 45.7% (95% CrI: 28.4–69.0%) of the population are estimated to have been infected (Fig. 2f). This estimate includes all infections and thus is much higher than the reported number of cases (3.77% of the population; see estimated infection-detection rates in Supplementary Fig. 3e). In addition, unlike the UK and South Africa where the pandemic wave rose and fell quickly, pandemic activity—based on national incidence and mortality curves—remained at high levels for a much longer duration (Fig. 2e). This may be due to the larger geographical area of Brazil, the aggregative nature of country-level incidence and

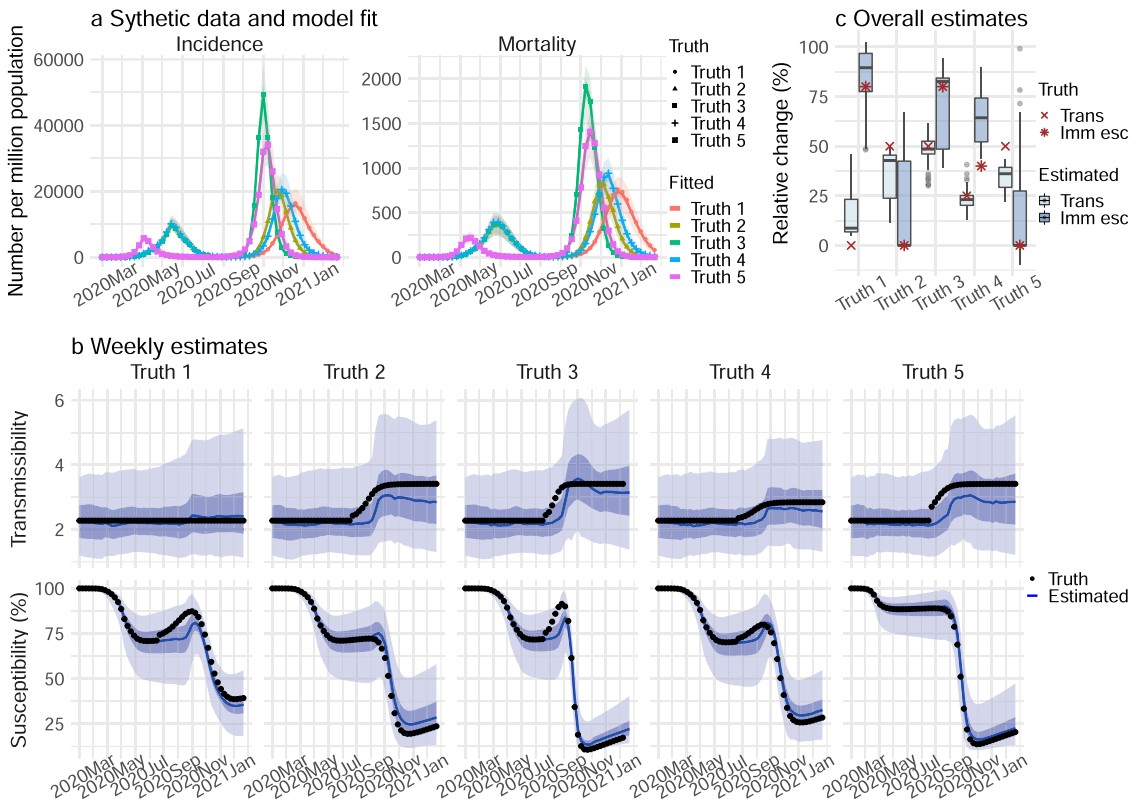

**Fig. 1 Model-inference system validation using model-generated synthetic data with an infection-detection rate of 20%.** For this testing, the true values of incidence and mortality by week (**a**), transmissibility by week (**b**, top panel), population susceptibility by week (**b**, bottom), and overall changes in transmissibility and immune escape due to a new variant (**c**) were generated by model simulations with prescribed parameters and conditions. Unlike the real world, in which most epidemiological characteristics are unobserved, here these quantities (i.e., the "Truth") are prescribed and known and thus can be compared to estimates made with the model-inference system using the synthetic, model-generated incidence and mortality data (**a**). **a** Five sets of synthetic data (dots) and model-fits to each data set; lines show mean estimates and surrounding areas show 50% (dark) and 95% (light) CrIs. **b** Weekly model-estimated transmissibility and population susceptibility. The lines show mean estimates and surrounding areas show 50% (dark) and 95% (light) CrIs, compared to the true values (dots). **c** overall estimates of the change in transmissibility (Trans) and immune escape (Imm esc), compared to the true values (dots); boxes show model-estimated median (middle bar) and interquartile range (box edges), and whiskers show model-estimated 95% CIs, from $n = 100$ model-inference simulations.

**Table 1 Estimated changes in transmissibility and level of immune evasion, compared to the wild-type virus.**

| Location | Variant | Changes in transmissibility (%) | Immune evasion (%) |
|---|---|---|---|
| United Kingdom | B.1.1.7 | 46.6 (32.3, 54.6) | 3.9 (0, 36.2) |
| South Africa | B.1.351 | 32.4 (14.6, 48) | 61.3 (42.6, 85.8) |
| Brazil | P.1 | 43.3 (30.3, 65.3) | 52.5 (0, 75.8) |

Numbers show model-estimated mean (95% CI) from 100 model-inference runs totaling 50,000 model realizations.

mortality data combining multiple outbreak waves from different sub-regions of the country, and the lack of national restrictions to curb the pandemic. Despite this large first pandemic wave, the emergence of the P.1 variant led to a second large pandemic wave from Dec 2020 onwards. Similar traveling waves through the country were evident from the incidence curve (Fig. 2e). By the week of 4/11/21, an additional 60.7% (95% CrI: 40.5–92.0%) of the population are estimated to have been infected, including re-infections.

**Estimated increased transmissibility and immune evasion.** Accounting for concurrent NPIs, vaccination, and seasonal

transmission trends (Supplementary Fig. 2), the model-inference system estimates also enable assessment of key properties specific to the three variants. Estimated transmissibility increased in conjunction with the widespread presence of the new variant in each country (Fig. 3b for B.1.1.7, Fig. 3e for B.1.351, and Fig. 3h for P.1). Overall, estimated viral transmissibility increased by 46.6% (95% CI: 32.3–54.5%) for the B.1.1.7 variant, 32.4% (95% CI: 14.6–48.0%) for the B.1.351 variant, and 43.3% (95% CI: 30.3–65.3%) for the P.1 variant, compared to the wild-type virus (Table 1). In addition, the model-inference system also detects large increases of population susceptibility for the B.1.351 and P.1 variants, but not the B.1.1.7 variant (Fig. 3f, i vs. c; and Table 1). Specifically, the model estimates immune evasion for the B.1.351 variant among 61.3% (95% CI: 42.6–85.8%) of the population infected with the wild-type virus during the first wave in South Africa and for the P.1 variant among 52.5% (95% CI: 0–75.8%) of the population infected with the wild-type virus during the first wave in Brazil.

**Competition among variants and potential future outcomes.** As many places have detected one or more VOCs locally, it is important to understand the potential pandemic outcomes given the characteristics of and competition among variants, interactions with ongoing NPIs, and mass-vaccination. We thus use a multi-variant, age-structured model to simulate potential

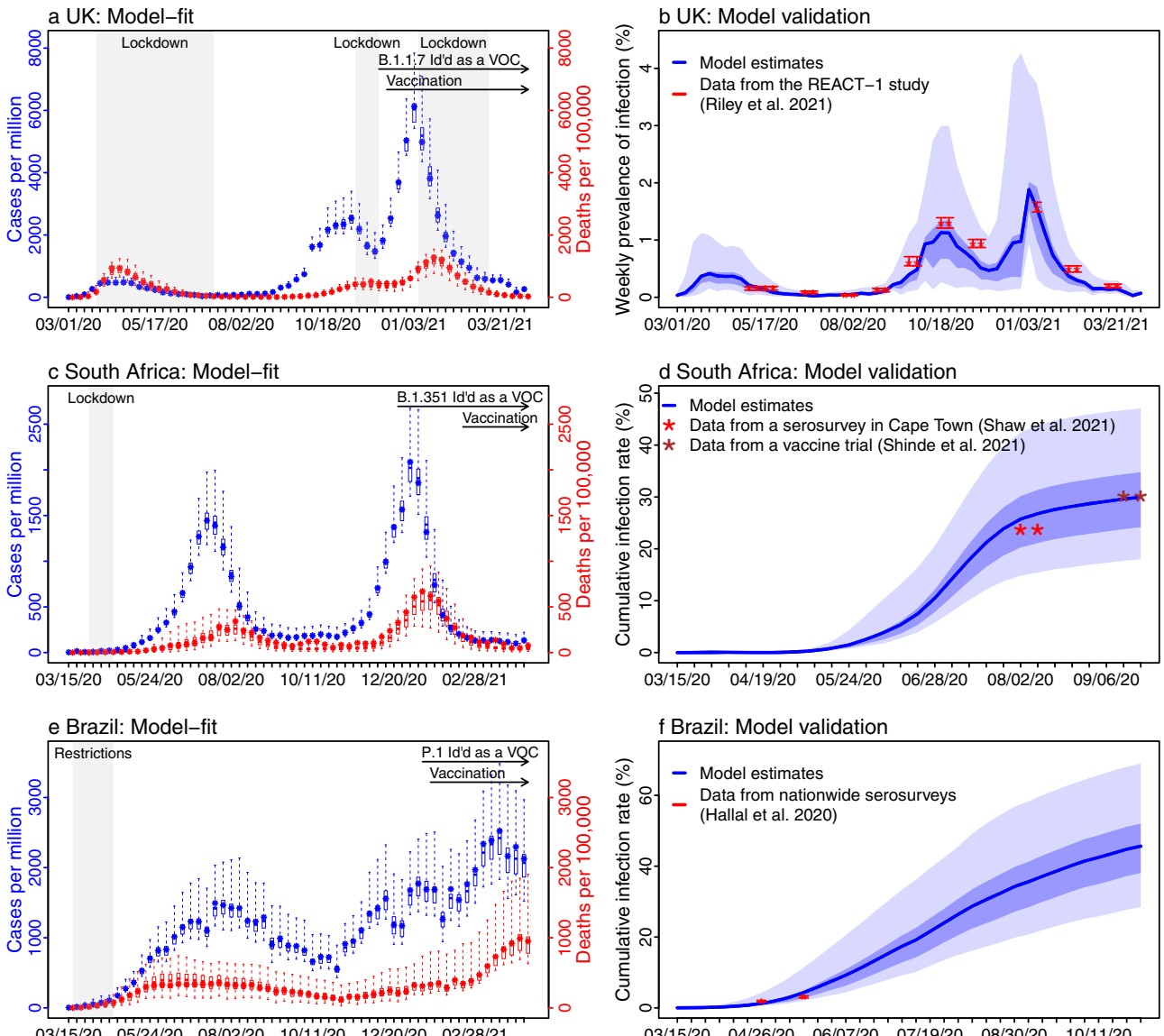

**Fig. 2 Model-inference system fit to data for the three countries and validation using independent datasets.** The left column shows the model fit to reported weekly case and mortality data for the UK (**a**), South Africa (**c**), and Brazil (**e**). Dots show the weekly number of cases (in blue) and deaths (in red) per 1 million persons; boxes (middle bar = mean; edges = 50% CrIs) and whiskers (95% CrIs) show the corresponding model estimates. Gray shaded boxes indicate the timing of lockdowns or key periods of restricted activity; horizontal arrows indicate the timing of variant identification and vaccination rollout. The right column compares available, independent measurements to corresponding model estimates. Red dots and error bars show measured prevalence over 10 periods of time from the REACT-1 study for the UK (**b**), cumulative infection rates from two serology studies in South Africa (**d**), and cumulative infection rates from two nationwide household serosurveys in Brazil (**f**). Blue lines and surrounding areas show model-estimated mean, 50% (dark) and 95% (light) CrIs. Model estimates (mean, 50% and 95% CrIs) are summarized over $n = 100$ model-inference runs (500 model replica each, totaling 50,000 model realizations).

pandemic outcomes for the period from May 2021 to Aug 2021 under scenarios with different variant prevalence, NPIs, and vaccine efficacy.

We focus on NYC where detailed data and estimates (e.g., contact patterns, variant prevalence, and vaccination rates) are available. We consider three variant prevalence scenarios for P.1 and B.1.351; these are equal initial seeding of P.1 and B.1.351 (both at 2–5%) and higher prevalence of either P.1 or B.1.351 (one at 2–5% and the other at 0.2–0.6%; Supplementary Table 5). For each variant prevalence scenario, we further consider 5 NPI scenarios and 4 VE reduction scenarios (20 combinations in total for each). The NPI scenarios range from maintaining the NPIs implemented at the start of a simulation to gradually lifting all NPIs within a 2-month

window. Lastly, because NYC mostly used mRNA vaccines during the study period, we use mRNA vaccines (here assumed 85%/95% for the 1st/2nd dose against the wild-type) as a baseline and scale the VEs to represent potential VE reduction against the new variants, ranging from no reduction to a large reduction (as low as 42.5%/57% for the 1st/2nd dose, i.e., a 50%/40% reduction for the 1st/2nd dose). In addition, because different types of vaccines with different VEs have been used elsewhere, to represent these diverse vaccines and VEs, we also simulate VEs ranging from very high (85%/95% for 1st/ 2nd dose) to high (60%/70%) and medium (45%/55%; Supplementary Fig. 5) levels.

Figure 4a and Supplementary Fig. 4 show example projections of infections and mortality assuming a best-case scenario in which

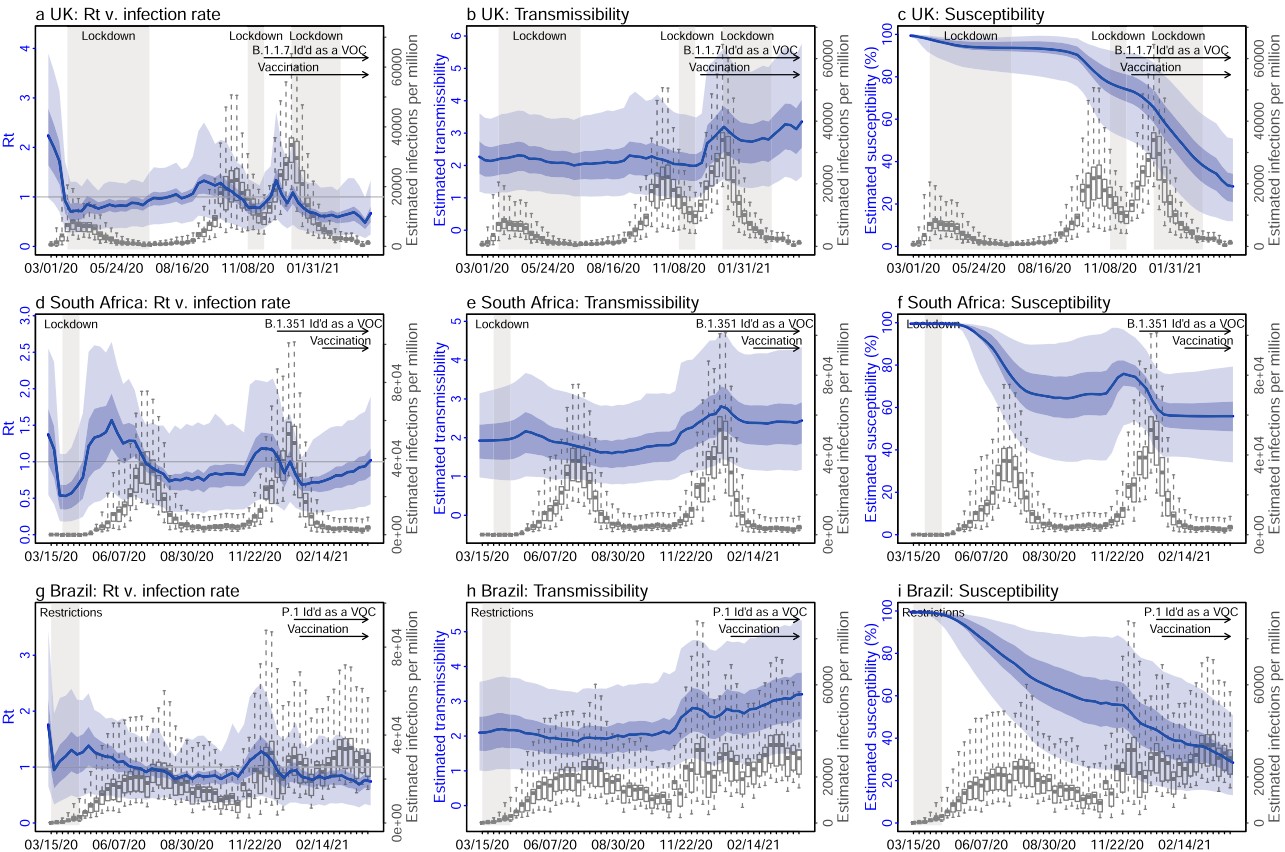

**Fig. 3 Key model-inference system estimates.** Left column (**a**, **d**, **g**) shows estimated real-time reproduction number $R_t$, middle column (**b**, **e**, **h**) shows estimated transmissibility, and right column (**c**, **f**, **i**) shows estimated population susceptibility for each week during the study period, for the three countries. For comparison, estimated weekly infection rates are superimposed in each plot (right y axis). Blue lines and surrounding areas show the estimated mean, 50% (dark) and 95% (light) CrIs. Boxes (middle bar = mean; edges = 50% CrIs) and whiskers (=95% CrIs) show the estimated weekly infection rates. Gray shaded areas indicate the timing of lockdowns or key periods of restricted activity; horizontal arrows indicate the timing of variant identification and vaccination rollout. Note that the transmissibility estimates (**b**, **e**, **h**) have removed the effects of changing population susceptibility, NPIs, and disease seasonality; thus, the trends are more stable than the reproduction number ($R_t$; **a**, **d**, **g**) and reflect changes in variant-specific properties. Model estimates (mean, 50% and 95% CrIs) are summarized over $n = 100$ model-inference runs (500 model replica each, totaling 50,000 model realizations).

mRNA vaccine-induced immunity is as effective against all three VOCs as for the wild-type virus. At the time of these simulations (i.e., end of April 2021), the B.1.1.7 variant was the predominant VOC in NYC; however, given their estimated propensity for immune escape, both B.1.351 and P.1 could outcompete B.1.1.7 and become predominant in the coming months (Fig. 4a, top panel). The relative prevalence of B.1.351 and P.1 depends largely on their initial introduction and establishment in the population. These two variants would arise at similar rates and co-dominate, if they are introduced and established in the population simultaneously (Fig. 4a, left panel). However, should either be established in the population ahead of the other, it would become dominant and suppress but not preclude the rise of the other variant (Fig. 4a, middle and right panels). In addition, the B.1.351 variant would be slightly more competitive if the vaccines are less effective against it than the P.1. variant (Fig. 4b and Supplementary Table 1), as has been shown in laboratory studies[7,9].

Tallies of model-projected infections (Fig. 4b and Supplementary Fig. 5b) and deaths (Supplementary Fig. 4) reveal four key determinants of future pandemic outcomes. First, simultaneous introduction of both the B.1.351 and P.1 variants would lead to larger increases of infections and mortality than the sole introduction of either variant (Fig. 4b and Supplementary Fig. 5b, Supplementary Fig. 4b). This result indicates the importance of limiting the introduction of multiple VOCs. Second, maintaining

very high vaccine efficacies against all variants is critical for mitigating the risk of a large resurgence in populations with relatively high vaccination coverage (e.g., compare the first four subplots in Fig. 4b). Third, continued non-pharmaceutical preventive measures will reduce infection resurgence as municipalities reopen economies. For instance, even with high vaccine efficacy, a rapid, full reopening before a very large proportion of the population is fully vaccinated could lead to approximately three times as many infections as when reopening occurs more slowly (Fig. 4a, b, first subplot). Maintaining NPIs is even more important in places where less efficacious vaccines are administered (Supplementary Fig. 5b). Four, reassuringly, while projected trends for COVID-19-related mortality are in general similar to those for infection (Supplementary Fig. 4a vs. Fig. 4a), lower proportions of COVID-19-related deaths would occur due to the increased transmission of B.1.351 and/or P.1 (Supplementary Fig. 4b vs. Fig. 4b; note the larger proportion of deaths due to B.1.1.7 than infections). This is due to the currently higher vaccination coverage among older adults who have been prioritized for vaccination in NYC similar to many other municipalities (see Supplementary Table 4 for vaccination coverage by age group). This finding emphasizes the importance of vaccine effectiveness against VOCs and of prioritizing vulnerable populations for vaccination in order to prevent severe outcomes of COVID-19.

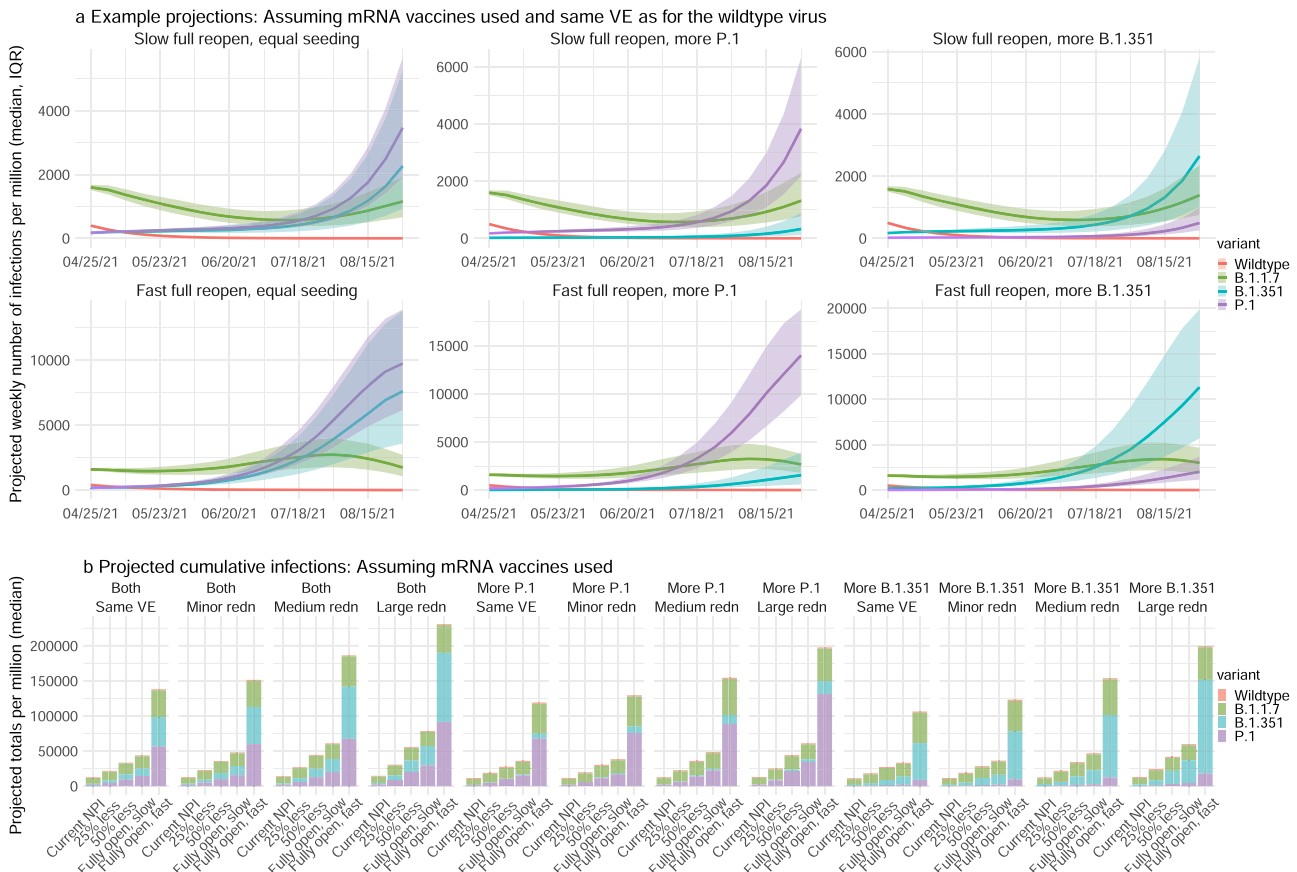

**Fig. 4 Model projections of infection, under different scenarios of VOC co-circulation, NPIs, and reduction of vaccine efficacy (VE) against the VOCs.**
**a** Example projected epidemic trajectories for each variant assuming mRNA vaccines are used and the VE is as high as for the wild-type virus. Lines and surrounding areas show model-projected median and interquartile range, from n = 1000 simulations (color-coded for each variant as indicated by the legend). **b** Tallies over the entire simulation period (May–Aug 2021) for different scenarios of seeding, change in VE (as indicated in the subtitles, see detail in Supplementary Table 5), and NPIs (as indicated by the x axis labels, see detail in Supplementary Table 5). All numbers are scaled to per 1 million people. For the projected percentages for each variant and uncertainty bounds, see Supplementary Table 1.

## Discussion

Despite vaccine availability, the future trajectory of the COVID-19 pandemic remains uncertain due to the potential additional emergence and continued spread of multiple VOCs. To improve understanding of future pandemic dynamics, here we have developed and applied a comprehensive model-inference system to quantify key viral properties for three VOCs: B.1.1.7, B.1.351, and P.1. Our estimates for the B.1.1.7 variant are consistent with detailed epidemiological evidence (32.5–54.6% increase in transmissibility and minimal immune evasion estimated here vs. 30–50% increase in secondary attack rate based on contact tracing data[3,4] and little immune evasion based on laboratory and real-world vaccination data[5,6,15]). Our estimates of the level of immune evasion for the B.1.351 and P.1 variants are also consistent with antibody neutralization data suggesting both variants can evade prior immunity induced by infection and vaccination, though to a larger extent for the B.1.351 variant[7,9]. Here we provide joint quantification of immune escape and the change in transmissibility for both variants. Overall, the model-inference system estimates and model simulations suggest that both B.1.351 and P.1 are likely more competitive than the B.1.1.7 variant due to their greater propensity for immune escape. These estimates are consistent with observations from Qatar[15] and Canada[16] showing that the proportion of infections caused by B.1.351 and/or P.1 increased despite earlier introduction and dominance of B.1.1.7 in these populations. Therefore, in spite of the current widespread prevalence of B.1.1.7 in Europe and North America, importation of B.1.351 and/or P.1 to these regions could replace B.1.1.7 dominance and lead to a further increase of infections by either B.1.351, P.1 or both variants. Mass-vaccination with highly effective vaccines is thus crucial for mitigating the risk of future SARS-CoV-2 resurgence, particularly as economies reopen.

Our model-inference system estimates substantial immune escape for both P.1 and B.1.351. For P.1, the mean estimate is bounded by a very broad confidence interval; however, for B.1.351 the uncertainty is more constrained and indicates greater confidence that a substantial level of immune escape occurs. These latter findings are supported by vaccine clinical trial and real-world data. In particular, Shinde et al.[13] found a similar likelihood of COVID-19, mostly due to B.1.351, among trial participants who were seropositive at enrollment (i.e., after the first wave) compared to those seronegative. Although potential differences in risk of exposure among the two comparison groups may have somewhat biased this finding, substantial repeat and breakthrough infection occurred due to B.1.351. Further, recent data from Qatar[15] indicate that individuals receiving the full dosing regimen of the Pfizer BNT162b2 mRNA vaccine are at greater risk of breakthrough infection with the B.1.351 variant than B.1.1.7. Continued monitoring of the severity of both repeat and breakthrough infections is needed to more fully understand the ongoing risks of VOCs to both health and the economy. In addition, the Qatar study[15] highlights the importance of full vaccination (i.e., administration of both doses for mRNA vaccines), as participants gained little protection against severe illness

after the first vaccine dose (vs. ~50% efficacy for B.1.1.7), even though full protection against severe illness was retained after two vaccine doses. It is thus likely critical that best vaccination protocols are followed in order to confer protection against variants with immune escape properties and that potential waning of vaccine-induced immunity over time is monitored.

In light of the spatial expansion of B.1.1.7, B.1.351, and P.1 and the potential emergence of other new variants, vaccination is paramount for controlling the COVID-19 pandemic. It is imperative that vaccine production, distribution, and administration proceed expeditiously, particularly in resource-limited settings. Without effective global control of the pandemic, the continued transmission of SARS-CoV-2 will give rise to additional new variants and pose new threats to all. As vaccines are distributed and administered, a continuation of non-pharmaceutical preventive measures is needed to minimize infections among the unvaccinated. As shown in our simulations, despite the relatively high vaccine coverage obtained to date in a place like NYC, COVID-19 infections could resurge if such locations lift preventive measures prematurely.

Due to a lack of sub-regional data, we used aggregated country-level data to estimate the properties for the three VOCs. Our model-inference system also did not account for differences in disease severity and infection-detection rate among age groups, which may vary substantially[17]. This model simplification may introduce uncertainty and bias to our estimates, particularly for Brazil where country-level data may mask more intense transmission in subpopulations and may have led to an underestimation of the transmissibility of P.1. Nevertheless, validation using independent data, including for Brazil, indicates that the model-inference system is able to closely capture pandemic dynamics and accurately estimate cumulative infection rates (Fig. 2). Further, because the model-inference system simultaneously accounts for population susceptibility, disease seasonality, NPIs, and vaccination, it is able to specifically estimate changes related to a given new variant and closely matches available epidemiological data (e.g., for B.1.1.7). The model simulations using these estimated characteristics further illustrate the relative competitiveness of the three VOCs and delineate key determinants of future infection outcomes. Overall, our findings point to the importance of preventing the spread of B.1.1.7, B.1.351, P.1, and other emerging VOCs or variants of interest (VOIs), via continued NPIs, prompt mass-vaccination of all populations, continued monitoring of vaccine efficacy, and potentially updating vaccine formulations to ensure high efficacy.

## Methods

**Data sources and processing**. The model-inference system uses reported COVID-19 case and mortality data to capture transmission dynamics, weather data to estimate disease seasonality, mobility data to represent concurrent NPIs, and vaccination data to account for changes in population susceptibility due to vaccination, for each of the three countries (i.e., the UK, South Africa, and Brazil). Country-level daily COVID-19 case and mortality data came from COVID-19 Data Repository of the Center for Systems Science and Engineering (CSSE) at Johns Hopkins University[18,19]; we aggregated the data to weekly intervals until the week of 4/11/2021 but excluded initial weeks with low case rates (<2 per million population). Hourly surface station temperature and relative humidity came from the Integrated Surface Dataset (ISD) maintained by the National Oceanic and Atmospheric Administration (NOAA) and are accessible using the "stationaRy" R package (version 0.5.1)[20]. We computed specific humidity using temperature and relative humidity per the Clausius–Clapeyron equation[21]. We then aggregated these data for all weather stations in each country with measurements since 2000 and calculated the average for each week of the year during 2000–2020. To compute the seasonal trend, we used a method developed by Yuan et al.[22], which estimates the relative reproduction number based on temperature and specific humidity (see details in Supplementary Methods). Daily mobility data were derived from Google Community Mobility Reports[23]; we aggregated all business-related categories (i.e., retail and recreational, transit stations, and workplaces) in all locations in each country to weekly intervals. Daily vaccination data (for 1st and

2nd dose, if applied) for the UK were sourced from Public Health England[24]; and data for South Africa and Brazil were obtained from Our World in Data[25,26].

**Model-inference system**. Contact tracing data capturing chains of transmission can be used to compute the secondary attack rate and quantify changes in transmissibility due to a given new variant. Surveillance data and laboratory viral characterization can be used to document and quantify levels of immune evasion. Yet such detailed data are often not available, particularly for resource-limited settings. Given these circumstances, mathematical modeling that assimilates epidemic surveillance data provides an attractive alternate means for estimating key epidemiological properties of novel variants, including the transmission rate. However, joint estimation of the transmission rate and population susceptibility, which is related to immune evasion, is challenging, as both quantities can change for a new variant. This problem arises mathematically because the product of these two quantities, rather than either individually, determines disease incidence—i.e., at any point in time, given the incidence, transmissibility, and susceptibility are not individually identifiable. Nevertheless, we posit that transmissibility and susceptibility affect epidemic dynamics differentially over time and, as such, data at multiple time points can enable joint estimation. Indeed, simulations show that epidemic trajectories can diverge over time even when infection rates during the initial weeks after the introduction of a new variant are similar (see, e.g., Fig. 1a). We thus design a model-inference system to estimate the most plausible joint changes in these two quantities using commonly available incidence and mortality time series.

The model-inference system is comprised of an epidemic model for simulating the transmission dynamics of SARS-CoV-2 and a statistical inference method for estimating the model state variables and parameters. The epidemic model is a susceptible-exposed-infectious-recovered-susceptible (SEIRS) construct that further accounts for two-dose vaccination. In addition, to include the effects of public health interventions and disease seasonality, it further adjusts the transmission rate each week using mobility data and the estimated seasonal trend based on climate conditions (see Supplementary Eq. S3 in the Supplementary Methods). The system then combines the model-simulated (prior) estimates and observed case and mortality data to compute posterior estimates using the ensemble Kalman adjustment filter (EAKF)[27]. We also apply a technique termed space re-probing[28] that accommodates possible large changes mid-pandemic to transmissibility and population susceptibility. Further, due to the challenge of identifying these two quantities individually, we ran the model-inference system, repeatedly and in turn, in order to test 14 major combinations of changes in transmissibility and susceptibility (see details in Supplementary Methods). Briefly, depending on the hypothesized change, we restricted the EAKF update to a given related set of parameters or variables. For instance, for the hypothesis that the new variant changes the transmissibility but does not evade immunity, the system only allows major adjustment to the transmission rate and the infectious period; for the hypothesis that the new variant induces both changes, the system allows major adjustment to the transmission rate, the infectious period, and population susceptibility. The system then selects the run with the best performance based on the accuracy of model-fit, one-step-ahead prediction, and magnitude of changes to key state variables to identify the most plausible combination of changes in transmissibility and level of immune evasion (see Supplementary Methods and Supplementary Fig. 6 for model goodness-of-fit measures). To approximate the distribution of the system (including all model state variables and parameters), we employed an ensemble of a model replica ($n = 500$ here) and updated the ensemble posterior each week using the EAKF. In addition, to account for model stochasticity, we repeated each model-inference simulation 100 times for each data set, each with initial parameters and variables are randomly drawn from prior distributions (Supplementary Table 2). Consequently, model estimates are aggregated from 50,000 model runs in total.

**Estimation of variant-specific changes in transmissibility and level of immune evasion**. The model decouples the impact of concurrent NPIs and disease seasonality from the transmission rate and infectious period (Supplementary Eq. S3); as such, estimates for the latter two parameters are variant-specific. We thus compute transmissibility as the product of the transmission rate and infectious period. To reduce uncertainty, we average transmissibility estimates over the first pandemic wave (excluding the first 3 weeks when model estimates are less accurate) for the wild-type SARS-CoV-2 virus; similarly, we average the transmissibility estimates over the period when the new variant is dominant. We identify this latter time period based on the transmissibility estimates: (1) For the UK (B.1.1.7) and South Africa (B.1.351), the estimated transmissibility increased and plateaued within 10 weeks (Fig. 3); we thus used the period from the week with the maximal transmissibility during the 10 weeks following its initial increase to the end of our study period (i.e., the week of 4/11/2021). However, for the UK, we excluded the 3rd lockdown period when estimated transmissibility is lower, potentially due to better awareness of B.1.1.7 and preventive measures taken at the time not fully captured by the model. Of note, contact tracing data also indicate a lower increase of the secondary attack rate around that time: 25-40% during 11/30/20–1/10/21 (among 1,364,301 cases for this expanded analysis)[4] vs. 30–50% during 11/30–12/20/20 (among 386,805 cases)[3]. (2) For Brazil (P.1), the estimated transmissibility increased more gradually (Fig. 3), we thus instead used either the weeks identified per 1 or the last 8 weeks of our study period, whichever with a longer time period, to ensure the robustness of estimation. We then compute the average change in

transmissibility due to a new variant as the ratio of the two averaged estimates (i.e., after: before the rise of the new variant).

To quantify immune evasion, we record all time points inducing major EAKF adjustments to posterior estimates of susceptibility, compute the change in immunity as $\Delta Imm = S_{t+1} - S_t + i_t$ (with $S_t$ as the susceptibility at time-$t$ and $i_t$ as the new infections occurring at time-$t$), and sum over all $\Delta Imm$ estimates to compute the total change in immunity due to the new variant. We then compute the level of immune evasion as the ratio of the total change in immunity during the second wave to the model-estimated population immunity at the end of the first wave. This ratio provides an estimate of the fraction of individuals previously infected who are susceptible to re-infection with the new variant.

For both quantities, we report the mean and 95% CI based on the mean estimates from 100 repeated model-inference runs.

**Model validation using model-generated synthetic data**. To test the accuracy of the model-inference system, we generated 10 synthetic datasets using a two-variant SEIRS model (Supplementary Eqn. S4) and different scenarios of changing transmissibility and immune evasion (Supplementary Table 3). In each scenario, a new variant was introduced at week 21 of the simulation. We then combined the incidence and mortality due to both variants and added noise drawn from a Poisson distribution to represent the observational error. We then applied the model-inference system to estimate the model variables and parameters for each synthetic data set, per the procedure described above for real data. For comparison with model-inference system estimates, we computed the true values of population susceptibility and transmissibility over time as the weighted average of the two variants based on the relative prevalence at each time point (i.e., each week).

**Model validation using independent data**. To compare model estimates with independent observations not assimilated into the model-inference system, we identified four relevant datasets: (1) the REACT-1 study, which measures the prevalence of SARS-CoV-2 using PCR-testing of volunteers from the general public living in the UK. At the time of this study, the REACT-1 study has conducted 10 rounds of testing during 5/1/2020–3/30/2021 ($n = 1,572,951$ tests in total)[10,11]; Fig. 2b plots our estimates of the prevalence of SARS-CoV-2 each week, overlaying all 10 measures from the REACT-1 study for corresponding time periods; (2) a serosurvey of workers in Cape Town, South Africa, conducted during 8/17–9/4/2020 ($n = 405$ participants)[12]; (3) serology tests among participants enrolled during 8/17–11/25/2020 in the Novavax NVX-CoV2373 vaccine phase 2a-b trial in South Africa ($n = 1324$)[13]. Given this long enrollment period, we used the centered 2-week window (9/29–10/13/20) to match with our model estimates. and (4) two nationwide random household serosurveys conducted in Brazil during 5/14–5/21/2020 ($n = 25,025$ participants) and 6/4–6/7/2020 ($n = 31,165$ participants)[14]. To account for the delay in antibody generation, we shifted the timing of each serosurvey by 14 days when comparing to model-inference system estimates of cumulative infection rates in Fig. 2d, f.

**Model simulations testing the relative competitiveness of VOCs and projecting future transmission dynamics**. Here we modified a multi-variant model previously developed for influenza virus[29] to include age structure and interactions (Supplementary Eq. S5). The multi-variant model accounts for: (1) competition between each pair of SARS-CoV-2 variants (e.g., wild-type and B.1.1.7) via cross-protective immunity; (2) variant-specific transmissibility and population susceptibility, based on estimates derived in this study; (3) variant-specific vaccine efficacy under different scenarios (Supplementary Table 5); (4) age-specific differences in vaccination coverage at the start of simulation and vaccination uptake rates for the simulation period (Supplementary Table 4); (5) seasonality; and (6) changes in NPIs under different scenarios (Supplementary Table 5). We used data from NYC for baseline vaccination coverage[30] and initial prevalence of different variants[31], as well as key model estimates (e.g., transmission rates and infection-fatality risk by age group; see Supplementary Table 5)[17,32,33]. As in the previous work[17,32,33], we included 8 age groups (i.e., <1, 1–4, 5–14, 15–24, 25–44, 45–64, 65–74, and 75+ year-olds) in the model to account for age-specific differences. To focus on the three VOCs, we only included the B.1.1.7, B.1.351, and P.1 variants and attributed all other variants as "wild-type" virus, even though at the start of the simulations, the B.1.526 variant made up approximately one-third of sequenced infections (N.b., the B.1.526 variant likely emerged locally in NYC; we estimated a ~20% increase in transmissibility and nominal immune evasion for this variant[34]; based on these estimates the impact of this variant is expected to be relatively minor). We did not account for potential differences in infection-fatality risk by variant, as such information is not available; therefore, the simulated mortality under different scenarios only reflects the relative infection rate by age group, for which we apply age-specific infection-fatality risk (Supplementary Table 5). In addition, due to uncertainty vis-à-vis the severity and infection-fatality risk among breakthrough infections (i.e., those who have been vaccinated), we only show mortality-related simulations for the "Same VE" scenario which assumes no reduction in VE.

**Reporting summary**. Further information on research design is available in the Nature Research Reporting Summary linked to this article.

## Data availability

All data used in this study are publicly available as described in the "Data sources and processing" section. Compiled datasets used in this study are available at https://github.com/wan-yang/covid_voc_study.

## Code availability

All source code and data necessary for the replication of our results and figures are made publicly available at https://github.com/wan-yang/covid_voc_study.

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

## Acknowledgements

This study was supported by the National Institute of Allergy and Infectious Diseases (AI145883), the National Science Foundation Rapid Response Research Program (RAPID; DMS-2027369) and a gift from the Morris-Singer Foundation.

## Author contributions

W.Y. designed the study (main), conducted the model analyses, interpreted results, and wrote the first draft. J.S. designed the study (supporting), interpreted results, and critically revised the manuscript.

## Competing interests

J.S. and Columbia University disclose partial ownership of SK Analytics. J.S. discloses consulting for BNI. W.Y. declares no competing interest.
