## [Peer Review File · Nature Communications]

REVIEWER COMMENTS

Reviewer #1 (Remarks to the Author):

This is a timely topic, as we are now at the stage of the COVID-19 pandemic where in most countries the most prevalent variant is no longer the wild type.

So understanding transmissibility and immune landscape is crucial, as we continue attempt mitigation of SARS-CoV-2.

While this is a interesting ms, there are quite a few issues that I had, revolving the modeling framework and most importantly how it is explained.

Specifically:

How do they account for under-detection This is not clear

Delays in reporting Why not have this varying?

Disease seasonality not clear how this was done. Why is there a fig S5 with climate variables?

NPIs not well explained (even with the tables in Si)

Also since different vaccines might better in terms of variant coverage I wonder if it worth while to have simulations where you have different vaccines? This is not clear in the NYC simulations

Not clear if the age structured model is calibrated to match NYC pop?

While the scenarios are comprehensive, the lack of detail leaves me unsure of the validity of these results.

More importantly, once these issues are addressed, I would suggest to simulate scenarios of different VE, even though we know some of the vaccines are highly effective on the VOCs.

There is not clarity on how the vaccination was introduced in the model.

Further, the incorporation mobility and seasonality is not well explained.

Aggregating data at the Country level data for these three countries is problematic, particularly because there is huge spatial heterogeneity. This is not really discussed

Fig 1B The whole truth concept is poorly explained

Fig 1C Consider decomposing this plot. It is very confusing and hard to examine.

Fig 4 suggested the wildtype is not longer circulating. Is these scenarios? Does this seem plausible??

FigS2 A consider adding vaccine uptake. In the Uk vaccination started in Dec 2020 and my End of March 2021 a considerable proportion of the population had at least received one dosage.

The figure legend needs a lot more detail.

Fig S4 Again this fig is really confusing. Needs more space between plots and a much more

comprehensive legend.

The age structure model is very poorly explained. Also while there is some inference done there is no reporting of goodness of fit (eg, some AIC or MLE)

Reviewer #2 (Remarks to the Author):

This manuscript described a model-inference framework to estimate changes in transmissibility and immune escape of new variants of SARS-CoV-2. I believe the research question is important, but I was not very sure how the inference could be performed with case and mortality data without stratifying by virus strains (i.e., by wild type and the new variant). Please see below for my specific comments.

1. The two-virus-type model was built mainly from Equation S4 in the Supplementary Information. However, it was not clear to me how the two viruses compete for the same susceptible population. What is the relationship between S_i , S_j , and the S_{ij} ? How many susceptible classes are there if a two-virus-type model is considered? Please clarify using the two-virus-type model as an example.

2. Figure 1: Please add legend or notation to Figure 1C for "transmissibility" and "immune escape".

3. Figure 2: Were any serology data against wildtype or new variant used in the model fitting for the UK? Also, except for the REACT data, were the S-dropout data used for the UK model fitting? What about the sequence data from UK? Will adding all these data improve the model fit?

4. Figure 2: The serology data of South Africa and Brazil were obtained before the emergence of VOC. Thus, one would imagine these serology data were essentially reflecting the transmission of the wildtype. Therefore, in these two countries, the inference of VOC transmissibility and immune escape was mainly based on case and death data. With no data about the percentage of wildtype and VOC over time for the second wave, was the model overfitting the data because there would be quite a lot of combination of possible values of transmissibility and immune escape that would result in the observed epidemic case and death data.

5. Methods: The ensemble Kalman adjustment filter (EAKF) was used in the parameter estimation rather than the more widely used likelihood-based Bayesian inference with Markov Chain Monte Carlo. Would you please explain what priors were used in EAKF and how the posterior estimates would be affected if the priors were changed.

REVIEWER COMMENTS

Reviewer #1 (Remarks to the Author):

This is a timely topic, as we are now at the stage of the COVID-19 pandemic where in most countries the most prevalent variant is no longer the wild type. So understanding transmissibility and immune landscape is crucial, as we continue attempt mitigation of SARS-CoV-2. While this is a interesting ms, there are quite a few issues that I had, revolving the modeling framework and most importantly how it is explained.

Specifically:

How do they account for under-detection This is not clear

- Under-detection of infection is accounted for in the observation model (see SI, Section 2.2, Lines 121 - 141). To make it clearer, we have now added a more descriptive header for the related section. Briefly, the model-inference system included a parameter for infection-detection rate (i.e., case ascertainment rate). During model-inference, the model-simulated infections (i.e. all infections including those not identified as cases) were lagged in time to account for delay in detection, and then further multiplied by the infection-detection rate to compute model-simulated cases. The model-simulated cases were then used as model prior estimate to compute the posterior along with the observed cases. The infection-detection rate and time-lag in detection are both estimated by the model-inference system along with other parameters and variables.

Delays in reporting Why not have this varying?

- We thank the reviewer for the suggestion. Indeed, we used a broad prior range for the delay in detection and allowed it to vary over time. Specifically, gamma distributions were used to model the time from infectiousness to detection (see SI, section 2.2). For each week, the mean and standard deviation (SD) of the distribution were estimated and updated by the model-inference system. Fig R1 below shows the estimates for the three countries. The distributions remained wide for all three countries, likely due to several factors. First, the data were not strong enough to support more constrained estimates for these parameters (i.e. the mean and the SD of the gamma distribution). Second, the infection demographics changed over time and thus estimates made for the entire population may mask the changes for key populations. For

instance, during more recent months, while delay in detection in older age groups may have decreased, more infections occurred in children and young adults, who tend to experience more mild or asymptomatic infection, and thus have lower detection rates and take a longer time to detect.

Fig R1. Posterior estimates of time from infectiousness to detection. Gamma distributions were used to model the time from infectiousness to detection (see SI, section 2.2). The mean and standard deviation (SD) of the distribution were estimated by the model-inference system. Left panel shows the posterior estimates for the mean of this distribution and right panel shows the estimates for the SD of this distribution, for each country. Thick lines and shaded areas show the mean, 50% credible intervals (darker blue), and 95% credible intervals (lighter blue).

Disease seasonality not clear how this was done. Why is there a fig S5 with climate variables?

- Estimation of disease seasonality was described in the SI, Section 1 (Lines 7 – 54). Two climate variables – i.e., temperature and specific humidity – were used to compute the seasonal trend. Therefore, we showed the climate variables in Fig S5 (i.e., Fig S7 in the revision). Briefly, the seasonality model (Eqn S1) assumed the basic reproductive number at a given week of the year [$R_0(t)$, a measure of how the likelihood of transmission changes over time due to seasonal factors] is modulated by both humidity and temperature during that week. For the humidity,

based on available studies for SARS-CoV-2, the model assumed both very low and very high humidity conditions are favorable for SARS-CoV-2 transmission and used a quadratic form (the first square bracket in Eqn S1) to represent this relationship. For temperature, the model assumed low temperatures promote transmission and temperatures above a certain threshold (i.e., T_c in Eqn. S1) limit transmission; this is represented by the terms in the second square bracket of Eqn. S1. Thus, given different inputs of humidity and temperature in different locations (shown in the revised Fig S7 for the three countries), the local seasonal trend can be estimated. To allow for model flexibility, we further scaled the weekly country output from Eqn S1 by the country mean output, such that this scaled output provides a *relative*, seasonality-related transmissibility for each week of the year.

The final results for the seasonal trends are shown in Fig S2 and used in the epidemic model to represent disease seasonality for each country via the parameter b_t in Eqn S3. For instance, for the UK (Fig S2A), the estimated seasonal trend b_t is 1.15 for week 1 (a week during winter), indicating the transmission rate for that week is 1.15 times the yearly average, due to the more favorable winter weather conditions; similarly, estimated b_t is 0.86 for week 27 (a week during summer), indicating the transmission rate for that week is 86% of the yearly average, due to the less favorable summer weather conditions.

To clarify these issues, we have added a note (SI, Lines 52-54) and the above example in the SI (Lines 74 - 80) and the legend of Fig S2.

NPIs not well explained (even with the tables in Si)

- Different NPIs (e.g. stay-at-home mandate for nonessential workers, school closures, masking mandate, and other social distancing rules and preventive measures) have been implemented during the COVID-19 pandemic, depending on study location and time period. In addition, the effectiveness of each NPI is unclear. It is thus difficult to detail and capture specific NPI effectiveness in the model. However, previous studies including our own have shown that the overall impact of NPIs (e.g., reduction in the reproduction number R_t) has been highly correlated with population mobility during the COVID-19 pandemic (see, e.g., Lasry et al. 2020; MMWR 69(15):451-7; Kraemer et al. 2020 Science. 368(6490):493-7. Yang et al. 2021 RSIF 18(175):20200822). Thus, here we used relative mobility, as observed in each study location, to represent the overall impact of NPIs. The mobility data used were described in the Methods of the main text, section "Data sources and processing" and model implementation was described in the SI, 2.1 Epidemic model. Briefly, the epidemic model (Eqn S3) included a variable m_t (relative mobility at week t) to include the observed mobility pattern (shown in Fig S2). For instance, for the UK (Fig S2A), relative mobility m_t was 41% of pre-COVID levels for the week starting March 22, 2020; per Eqn 3, the overall transmission rate for that week was reduced by 41%, *before* adjusting for the effectiveness. To further account for the effectiveness, the model included another parameter e_t (effectiveness of NPIs) and multiplied it with m_t . The effectiveness e_t was estimated by the model-inference system along with other model variables and parameters.

To clarify these issues better, we have now reorganized the descriptions in Section 2.1 to highlight each component. In addition, we have also added examples similar to the ones described above to provide more context in the revision. Please see SI, Lines 82 – 98.

Also since different vaccines might better in terms of variant coverage I wonder if it worth while to have simulations where you have different vaccines? This is not clear in the NYC simulations

- We thank the reviewer for the suggestion. For NYC, 97% of vaccine doses administered through the end of April, 2021 were the Pfizer-BNTech or Moderna mRNA vaccines, both are highly effective (VE set to 85% after the 1st dose and 95% after the 2nd dose). Therefore, for the different VE scenarios simulated, we scaled the reduction due to the new variants to the baseline 85% and 95% VEs (see Table S5). Among the VE scenarios tested, the lowest VE was 59.5% for the 1st dose and 76% for the 2nd dose, against B.1.351 (Table S5); these settings are close to the VE reported for the J&J vaccine but may be still higher than other vaccines. Therefore, to represent vaccines with lower VE, we have now added a fourth VE scenario (i.e. lower VE) where after the assumed reduction, the VE would be 42.5% and 57% for the 1st and 2nd vaccine dose, respectively.

In addition, per the reviewer suggestion, we have now performed additional simulations using settings representing three levels of VE for different vaccines currently in use: 1) 85% (1st dose) and 95% (2nd dose) as reported for the mRNA vaccines; 2) 60% (1st dose) and 70% (2nd dose) as reported for vaccines such as J&J; and 3) 45% (1st dose) and 55% (2nd dose) as reported for vaccines such as Sinovac.

In the revision, we have added a summary of the simulation scenarios, including the new scenarios for difference vaccines in the Results, main text (see below). In addition, we report the new results in Fig 4 and Fig S5 of the revised manuscript.

See Main text, Lines 153 – 167:

“We focus on NYC for which detailed data and estimates (e.g., contact patterns, variant prevalence, and vaccination rates) are available. We consider 3 variant prevalence scenarios for P.1 and B.1.351; these are equal initial seeding of P.1 and B.1.351 (both at 2-5%) and higher prevalence of either P.1 or B.1.351 (one at 2-5% and the other at 0.2-0.6%; Table S5). For each variant prevalence scenario, we further consider 5 NPI scenarios and 4 VE reduction scenarios (20 combinations in total for each). The NPI scenarios range from maintaining the NPIs implemented at the start of a simulation to gradually lifting all NPIs within a 2-month window. Lastly, because NYC mostly used mRNA vaccines during the study period, we use mRNA vaccines (here assumed 85%/95% for the 1st/2nd dose against the wildtype) as a baseline and scale the VEs to represent potential VE reduction against the new variants, ranging from no reduction to a large reduction (as low as 42.5%/57% for the 1st/2nd dose, i.e. a 50%/40% reduction for the 1st/2nd doses). In addition, because different types of vaccines with different VEs have been used elsewhere, to represent these diverse vaccines and VEs, we also simulated VEs ranging from very high (85%/95% for 1st/2nd dose) to high (60%/70%) and medium (45%/55%; see Fig S5) levels.”

Not clear if the age structured model is calibrated to match NYC pop?

- We used NYC data (e.g. population age structure and size) and model estimates to initialize the multiple-variant age-structured model and perform the simulations. We have worked closely with the NYC health department since March 2020 when NYC became the initial pandemic epicenter in the US (see modeling results at <https://github.com/wan-yang/COLUMBIA-COVID19-PROJECTIONS-FOR-NYC>) and have used detailed estimates to support model formulation and simulations. For this work, the objectives for the simulations are to 1) test the relative competitiveness of the three VOCs under different seeding, NPI, and vaccination scenarios; and 2) identify key factors determining the impact of the VOCs. While the simulations are calibrated to the NYC population, we summarize and report the general modeling findings, as we believe they are applicable to other municipalities, rather than specific to NYC.

While the scenarios are comprehensive, the lack of detail leaves me unsure of the validity of these results. More importantly, once these issues are addressed, I would suggest to simulate scenarios of different VE, even though we know some of the vaccines are highly effective on the VOCs.

- We thank the reviewer for the suggestions. We have revised the manuscript to provide additional methodological detail and clarification. As described above, we have also performed additional simulations for the vaccination scenarios, using settings representing three levels of VE (versus wild type), covering different vaccines currently in use.

There is not clarity on how the vaccination was introduced in the model.

- We have revised the Methods section to clarify the vaccination component in the model:

SI, Section 2.1, Lines 104 – 119:

“To incorporate changes in population susceptibility due to vaccination, the model accounts for two-dose vaccination via $v_1(t)$ and $v_2(t)$. Specifically, $v_1(t)$ is the number of individuals successfully immunized after the first dose of the vaccine and is computed using vaccination data and vaccine efficacy for one dose (see detailed settings in Table S2). Similarly, $v_2(t)$ is the additional number of individuals successfully immunized after the second vaccine dose (excluding those successfully immunized after the first dose). As an example, assume 100 individuals receive their 1st dose on day 0 and 2nd dose on day 21 and the vaccine efficacy (VE) is 85% 14 days after the 1st dose and 95% 7 days after the 2nd dose. To compute v_1 and v_2 , we first adjust for immunity gained from infection; here for illustration, assume 10% have immunity from prior infection at time of vaccination (for simplicity, assumed the same for both doses here; in this study, these were based on model-estimated cumulative infection rate and

adjusted for waning immunity), then the number of individuals susceptible at time of vaccination would be $100 \times (1 - 10\%) = 90$. Further adjusting for the time lag of immunity development and VE, $v_1(t = \text{day } 14)$ is calculated as $90 \times 85\% = 76.5$ and $v_2(t = \text{day } 28)$ is calculated as $90 \times (1 - 85\%)[1 - (1 - 85\%)(1 - 95\%)] = 9$, such that the total percentage of individuals successfully immunized is $(76.5 + 9) / 90 = 95\%$, i.e. VE for the fully vaccinated.”

Further, the incorporation mobility and seasonality is not well explained.

- As detailed above, we have revised the manuscript to clarify these modeling components.

Aggregating data at the Country level data for these three countries is problematic, particularly because there is huge spatial heterogeneity. This is not really discussed

- We thank the reviewer for the critique and have acknowledged this limitation in the Discussion:

Main text, Lines 260 -268:

“Due to a lack of sub-regional data, we used aggregated country-level data to estimate the properties for the three VOCs. Our model-inference system also did not account for differences of disease severity and infection-detection rate among age groups, which may vary substantially.¹⁷ This model simplification may introduce uncertainty and bias to our estimates, particularly for Brazil where country-level data may mask more intense transmission in subpopulations and may have led to underestimation of the transmissibility of P.1. Nevertheless, validation using independent data, including for Brazil, indicates that the model-inference system is able to closely capture pandemic dynamics and accurately estimate cumulative infection rates (Fig 2).”

Fig 1B The whole truth concept is poorly explained

- We have revised Fig 1 and its legend to better explain this concept. See the revised figure and legend below.

Fig 1. Model-inference system validation using model-generated synthetic data with an infection-detection rate of 20%. For this testing, the true values of incidence and mortality by week (A), transmissibility by week (B, top panel), population susceptibility by week (B, bottom), and overall changes in transmissibility and immune escape due to a new variant (C) were generated by model simulations with prescribed parameters and conditions. Unlike the real-world in which most epidemiological characteristics are unobserved, here these quantities (i.e. the ‘Truth’) are prescribed and known and thus can be compared to estimates made with the model-inference system using the synthetic, model-generated incidence and mortality data (A). (A) 5 sets of synthetic data (dots) and model-fits to each dataset; lines show mean estimates and surrounding areas show 50% (dark) and 95% (light) CRIs. (B) Weekly model estimated transmissibility and population susceptibility. The lines show mean estimates and surrounding areas show 50% (dark) and 95% (light) CRIs, compared to the true values (dots). (C) overall estimates of the change in transmissibility (‘Trans’) and immune escape (‘Imm esc’; boxes and whiskers show model estimated median, interquartile range, and 95% CI from 100 model-inference simulations) compared to the true values (dots).

Fig 1C Consider decomposing this plot. It is very confusing and hard to examine.

- We have revised Fig 1C and its legend to clarify the figure items.

Fig 4 suggested the wildtype is not longer circulating. Is these scenarios? Does this seem plausible??

- Yes, rapid variant displacement has occurred in all three countries (the UK, South Africa, and Brazil) where the three VOCs studied here initially emerged.

FigS2 A consider adding vaccine uptake. In the Uk vaccination started in Dec 2020 and my End of March 2021 a considerable proportion of the population had at least received one dosage. The figure legend needs a lot more detail.

- Per the reviewer suggestion, we have added vaccination uptake to Fig S2 and revised the legend to provide more detail. See the revised figure and legend below.

Fig S2. Pandemic dynamics, mobility, and estimated seasonal trends, and vaccination uptake in the three countries: (A) United Kingdom, (B) South Africa, and (C) Brazil. Bars show weekly

numbers of reported cases per 1 million persons (grey) and reported deaths per 100,000 persons (red). Green lines show relative mobility by week (relative to pre-pandemic levels), based on data from Google Community Mobility Reports (see main text, Methods, “Data sources and processing”). Orange lines show estimated seasonal trends (see SI, Section 1, “Estimating seasonal trends”). Blue lines show the cumulative fractions of population partially vaccinated (dashed lines) and fully vaccinated (solid lines), based on vaccination data from each country (see main text, Methods, “Data sources and processing”).

Fig S4 Again this fig is really confusing. Needs more space between plots and a much more comprehensive legend.

- We have revised the figure and legend to provide more detail. See the revised figure and legend below:

Fig S4. Model projections of COVID-19 related mortality under different scenarios of VOC co-circulation and NPIs. (A) Example projected weekly number of deaths due to infections by each variant, assuming mRNA vaccines are used and the vaccine efficacy (VE) is as high as for the

wild-type virus. Each panel shows projections for one seeding and NPI scenario, as indicated in the subtitle. Lines and surrounding areas show model projected median and interquartile range (color-coded for each variant as indicated by the legend). (B) Tallies over the entire simulation period (May – Aug 2021) for different scenarios of seeding (as indicated in the subtitles) and NPIs (as indicated by the x-axis labels). Due to the uncertainty of the infection fatality risk among breakthrough infections (i.e., those who have been vaccinated), all simulations shown here assume no reduction in VE. All numbers are scaled to per 1 million people.

The age structure model is very poorly explained. Also while there is some inference done there is no reporting of goodness of fit (eg, some AIC or MLE)

- We have added a description to explain the age-related component of the model. Please see SI, Lines 292– 296:

“...The formulation of the age-related model component follows a typical age-structured model; for a given age group A (designated by the superscript), the total infection rate by a given variant- j (designated by the subscript) is the sum over all age groups (i.e., $\sum_a \frac{\beta_j^{Aa} S_j^A I_j^a}{N^a}$, where β_j^{Aa} is the transmission rate of variant- j from age-group a to age-group A)...”

The ensemble adjustment Kalman filter (EAKF) inference method used here is a Bayesian method and does not generate AIC or MLE. However, we evaluate model performance based on multiple criteria, as described in SI, Section 2.3 “Inference using the EAKF” (see SI, Lines 223 – 238). To show these results, we have now added a Figure S6 to show two main criteria: 1) model fitting to case and mortality data, as indicated by the relative root-mean-squared-error (RRMSE) between the *posterior* estimates for the corresponding variable (i.e. case rate or mortality rate) and data; and 2) the accuracy of one-step ahead prediction, as indicated by the RRMSE between the *prior* estimates for the corresponding variable (i.e. case rate or mortality rate) and data.

Fig S6. Model goodness-of-fit. Boxes and whiskers show the distributions (median, interquartile range, and 95% CI across 100 model-inference runs) of relative root-mean-square-error (RRMSE) for the posterior estimates (cases in blue and deaths in pink) and one-step-ahead predictions. See SI, Section 2.3 for details on RRMSE calculation.

Reviewer #2 (Remarks to the Author):

This manuscript described a model-inference framework to estimate changes in transmissibility and immune escape of new variants of SARS-CoV-2. I believe the research question is important, but I was not very sure how the inference could be performed with case and mortality data without stratifying by virus strains (i.e., by wild type and the new variant). Please see below for my specific comments.

1. The two-virus-type model was built mainly from Equation S4 in the Supplementary Information. However, it was not clear to me how the two viruses compete for the same susceptible population. What is the relationship between S_i , S_j , and the S_{ij} ? How many susceptible classes are there if a two-virus-type model is considered? Please clarify using the two-virus-type model as an example.

- We thank the reviewer for the question. Previous models tend to track the infection history of different variants; for example, for a 2-virus system, depending on infection history, there would be 3 categories of susceptibility: S_i , S_j , and S_{ij} . Under this model construct, the number of categories exponentiates with the number of viruses in the system, making the system extremely high dimensional and complex for systems with more than 2 or 3 viruses.

Unlike such previous models, the multi-variant models in this study (Eqn S4 and Eqn S5) take a status-based construct, similar to developed in Gog and Grenfell (PNAS 2002) and Yang et al. (PLOS Comput Biol 2020). The basic idea is that rather than tracking infection history, the

model only tracks susceptibility to each variant. To do so, the model computes the change in susceptibility to variant- i as the sum of two components: 1) loss of susceptibility due to infection by the same variant and 2) loss of susceptibility due to infection by other variants via cross-immunity. In Eqn S4, c_{ij} represents the strength of immunity to variant- i conferred by infection with variant- j . For 1) where specific immunity is conferred, $c_{ii} = 1$ so that the loss of susceptibility is $c_{ii} \frac{\beta_i S_i I_i}{N} = \frac{\beta_i S_i I_i}{N}$, the same as in a simple SIR-type model (note, for illustration, we omit the terms $b_t e_t m_t$ here). For 2) where cross-immunity is conferred, $c_{ij} \in [0,1]$, the strength of cross-immunity to variant- i conferred by infection with variant- j depends on how closely related the pair of viruses i and j is. If the two variants are highly similar (e.g., for $i=B.1.1.7$ and $j=wildtype$), c_{ij} would be close to 1 (say, $c_{ij} = 0.95$) such that infection of $j=wildtype$ (i.e., $\frac{\beta_j S_j I_j}{N}$) confers 95% immunity protection against $i=B.1.1.7$ (i.e., $c_{ij} \frac{\beta_j S_j I_j}{N} = 0.95 \frac{\beta_j S_j I_j}{N}$). Conversely, if the two variants are highly dissimilar (e.g., for $i = B.1.351$ and $j = wildtype$), c_{ij} would be closer to 0 (say, $c_{ij} = 0.4$) such that infection of $j=wildtype$ (i.e., $\frac{\beta_j S_j I_j}{N}$) confers only 40% immunity protection against $i=B.1.351$ (i.e., $c_{ij} \frac{\beta_j S_j I_j}{N} = 0.4 \frac{\beta_j S_j I_j}{N}$). The summation in the RHS of line 1 of Eqn S4 (i.e., $\sum_j \frac{b_t e_t m_t c_{ij} \beta_j S_j I_j}{N}$) thus converts both specific immunity (for $i = j$) and cross-immunity (for $i \neq j$) to loss of susceptibility to each variant.

As shown above, using the status-based construct, the multi-variant model is greatly simplified and still able to represent both specific and cross-immunity and the resulting variant competition (For previous studies using this model construct, see, e.g., Koelle et al. 2006 Science; Koelle et al. 2015 eLife; and Arinaminpathy et al. 2012 PNAS, in addition to Gog and Grenfell 2002 PNAS 2002 and Yang et al. 2020 PLOS Comput Biol).

We have described how the cross-immunity settings and values of c_{ij} 's are configured in the SI (see Lines 321 – 329).

2. Figure 1: Please add legend or notation to Figure 1C for “transmissibility” and “immune escape”.

- We thank the reviewer for the suggestion and have added a more descriptive legend for Figure 1C and revised the Figure 1 caption. Please see the revised Figure 1 included in the reply to Reviewer 1 (page 7 of this document).

3. Figure 2: Were any serology data against wildtype or new variant used in the model fitting for the UK? Also, except for the REACT data, were the S-dropout data used for the UK model fitting? What about the sequence data from UK? Will adding all these data improve the model fit?

- For all three countries (i.e., including the UK), *only case and mortality data were used for model fitting*. The REACT data were independent of the model-inference estimates and only plotted to compare with corresponding model-estimates for model validation. Please note

that, while the UK has much richer epidemiological data and inclusion of these data could further improve model estimates, we develop the model-inference system to only require the minimal datasets commonly available for many countries (i.e. case and mortality data). As demonstrated here, despite the minimal data need, the model-inference system was able to accurately estimate observations not fed into system (e.g. the REACT data from the UK and serology data for South Africa and Brazil).

4. Figure 2: The serology data of South Africa and Brazil were obtained before the emergence of VOC. Thus, one would imagine these serology data were essentially reflecting the transmission of the wildtype. Therefore, in these two countries, the inference of VOC transmissibility and immune escape was mainly based on case and death data. With no data about the percentage of wildtype and VOC over time for the second wave, the model overfitting the data because there would be quite a lot of combination of possible values of transmissibility and immune escape that would result in the observed epidemic case and death data.

- As noted above, the serology data were not fed to the model-inference system; rather, they were only plotted to compare with corresponding model-estimates for model validation.

Indeed, because the infection rate is a product of transmissibility and susceptibility, multiple combinations of possible values of transmissibility and immune escape could lead to the same infection rate, *for a given time point*. Therefore, the key question here is two-fold: 1) Can the true values of transmissibility and immune escape be identified jointly? and 2) If yes, how can this be done? For the first question, when looking at different combinations of transmissibility and immune escape, we learned that over multiple points in time over the course of a pandemic wave, the epidemic trajectories diverge under different combinations, even though initially the infection rates could be similar. This is shown in Figure 1, where even though the five combinations give similar infection rates initially (from seeding of a new variant in ~late August to Sep/Oct), the epidemic trajectories diverge over time. For instance, for truth 1 (red line), the true change is 80% immune escape and no change in transmissibility; and for truth 2 (olive line), the true change is 0% immune escape and 50% increase in transmissibility; even with the higher percentage change in immunity, the increase in transmissibility leads to more immediate increase in infection rates (see the 2nd wave in olive due to a 50% increase in transmissibility is ahead of the one in red due to a 80% immune escape). These differences suggest that different combinations of changes in transmissibility and immune escape can be distinguished using observations over multiple time points. We have provided this rationale in the main text:

Main text, Lines 309 – 320:

“... However, joint estimation of the transmission rate and population susceptibility, which is related to immune evasion, is challenging, as both quantities can change for a new variant. This problem arises mathematically because the product of these two quantities, rather than either individually, determines disease incidence – i.e., at any point in time, given the incidence, transmissibility and susceptibility are not individually identifiable. Nevertheless, we posit that

transmissibility and susceptibility affect epidemic dynamics differentially over time, and, as such, data at multiple time points can enable joint estimation. Indeed, simulations show that epidemic trajectories can diverge over time even when infection rates during the initial weeks after introduction of a new variant are similar (see, e.g., Fig 1A). We thus design a model-inference system to estimate the most plausible joint changes in these two quantities using commonly available incidence and mortality time series...”

The answer to question #2 – how to estimate the two quantities jointly – is the main methodology advance in this study. We describe the method briefly in the main text (Lines 322 – 350 and excerpt below) and in detail in the SI for specific settings under the 14 hypothesized possible combinations and criteria for evaluating the hypothesis (See SI, Lines 143 – 246). As reported in the manuscript, we first tested this approach using model-synthesized data, where the true values for changes in transmissibility and immune escape due to the new variant, as well as the underlying state variables (e.g. population susceptibility), are known. As shown in Fig 1B, even though the combined transmissibility (black dots, top panel in Fig 1B) and population susceptibility (black dots, bottom panel in Fig 1B) could change substantially over time and that these underlying data were not fed into the model-inference system, the model-inference system was able to accurately estimate these underlying quantities (blue lines and shaded areas). This synthetic testing demonstrates the model-inference system is able to accurately estimate the changes in transmissibility and immune escape using only case and mortality data. In addition to this synthetic testing, when applied to real-world data from the three countries, we further validate the model estimates using independent data not fed into the model-inference system. These validation results are shown in Fig 2, B for the UK, D for South Africa, and F for Brazil.

Excerpt from Main text, Lines 332 – 344:

“...Further, due to the challenge identifying these two quantities individually, we ran the model-inference system, repeatedly and in turn, in order to test 14 major combinations of changes in transmissibility and susceptibility (see details in Supplemental Information). Briefly, depending on the hypothesized change, we restricted the EAKF update to a given related set of parameters or variables. For instance, for the hypothesis that the new variant changes the transmissibility but does not evade immunity, the system only allows major adjustment to the transmission rate and the infectious period; for the hypothesis that the new variant induces both changes, the system allows major adjustment to the transmission rate, the infectious period, and population susceptibility. The system then selects the run with the best performance based on the accuracy of model-fit, one-step ahead prediction, and magnitude of changes to key state variables to identify the most plausible combination of changes in transmissibility and level of immune evasion (see Supplemental Information and Fig S6 for model goodness-of-fit measures)...”

5.Methods: The ensemble Kalman adjustment filter (EAKF) was used in the parameter estimation rather than the more widely used likelihood-based Bayesian inference with Markov

Chain Monte Carlo. Would you please explain what priors were used in EAKF and how the posterior estimates would be affected if the priors were changed.

- The EAKF method has also been widely used in inference of infectious disease systems, including for COVID-19/SARS-CoV-2 (see, e.g., Yang et al. *The Lancet Infectious diseases*. 2021;21(2):203-12; Pei et al. *PNAS* 2018;115(11):2752-7; Shaman et al. *Nature Communications*. 2013; 4:2837). Here we use the EAKF because it is responsive to changes in the system, due to, e.g., the emergence of a new variant. This is important for the system studied here, because as shown in Fig 1B, even after a new variant is introduced, the parameters and state variables (e.g., transmissibility and susceptibility) do not change immediately to the characteristics of the new variant. Rather, the overall epidemic characteristics are a combination of the two viruses and will change gradually over time, depending on how quickly the new variant emerges and the relative prevalence of wildtype and the new variant. For instance, compare the changes in susceptibility for Truth 1 (80% immune escape and no change in transmissibility; 1st column, lower panel in Fig 1B) and Truth 3 (80% immune escape and 50% increase in transmissibility; 3rd column, lower panel in Fig 1B). Given the same level of immune escape, the increase in susceptibility in Truth 1 occurs more slowly than Truth 3, as Truth 1 does not include the additional 50% increase in transmissibility and thus the new variant in that scenario emerges more slowly. As the exact timing of new variant introduction and its transmissibility and immune escape property are all unknown, it is challenging for methods like MCMC to infer all these quantities jointly. However, as demonstrated in our study, the model-inference system using the EAKF and algorithm design here is able to infer the changes in underlying model parameters and state variables – after a few weeks adjusting to the new system dynamics, the model-inference estimates start to closely track the true values for both scenarios (again, please see comparison of the model-inference estimates in blue and the true values shown by black dots in Fig 1B).

Prior ranges for all parameters and state variables are reported in Table S2. In general, given the uncertainty, we use very wide prior ranges guided by the literature. For instance, a uniform prior of U[0.5, 0.8] per day is used for transmission rate (β) in the UK and a uniform prior of U[2, 5] days is used for the mean infectious period; these result in initial R_0 (for the wildtype) in the range of 1 to 4. However, even with these wide prior ranges, a substantial change beyond the specified prior range could occur for some parameter, due to a new variant (e.g. a 50% increase in transmission rate). Using the EAKF, the posterior parameter estimates can migrate outside initial prior ranges, which allows the model-inference system flexibility to capture such scenarios. For specific countries, in initial test-runs, we compared the posterior and prior estimates (the two should be similar after a few weeks of filter initial spinning) to fine tune some of the key parameters; these location-specific priors are all listed in Table S2.

REVIEWERS' COMMENTS

Reviewer #1 (Remarks to the Author):

I am happy with the changes made as per my comments/ suggestions. I also went through the comments from the other reviewer and believe these were addressed appropriately.

Reviewer #2 (Remarks to the Author):

The authors have largely addressed all my comments, but I still have a last question regarding the response to my Comment 4.

In real world, will the proposed model work better for scenarios of South Africa and Brazil where the wild type and the variant formed two distinct waves? Does the model require input about the date when the variant first emerged or was introduced into the population (i.e., similar to the UK scenario)? If no, how would the model be revised to detect the changes in epidemic growth rates due to the combined effect of increased transmissibility and immune escapes?

REVIEWERS' COMMENTS

Reviewer #1 (Remarks to the Author):

I am happy with the changes made as per my comments/ suggestions. I also went through the comments from the other reviewer and believe these were addressed appropriately.

--We thank the reviewer for the effort.

Reviewer #2 (Remarks to the Author):

The authors have largely addressed all my comments, but I still have a last question regarding the response to my Comment 4.

In real world, will the proposed model work better for scenarios of South Africa and Brazil where the wild type and the variant formed two distinct waves? Does the model require input about the date when the variant first emerged or was introduced into the population (i.e., similar to the UK scenario)? If no, how would the model be revised to detect the changes in epidemic growth rates due to the combined effect of increased transmissibility and immune escapes?

--We thank the reviewer for the questions. We do not observe a clear difference in model performance, related to the timing of variant emergence and whether the wild type and the variant formed two distinct waves. Please see Fig S6 for a comparison of model performance for the three countries. The model-inference system requires input of the rough timing of the ending of the first wave (which can be identified based on the incidence curve) but not the exact date when the variant first emerged or was introduced into the population. In addition to estimating the changes in transmissibility and immune escape separately, the model-inference system also estimates the effective reproduction number for each week (R_t). Similar to the epidemic growth rates, these R_t estimates reflect the combined effect of increased transmissibility and immune escape. These estimates are shown in Fig 3 (a for the UK, d for South Africa, and g for Brazil) and described in the Results (see section "Reconstructed pandemic dynamics in the three countries").